# Amoxicillin-Resistant Streptococci Carriage in the Mouths of Children: A Systematic Review and Meta-Analysis

**DOI:** 10.3390/pathogens11101114

**Published:** 2022-09-28

**Authors:** Ayrton G. Araújo Júnior, Marina L. V. A. Costa, Felipe R. P. Silva, Daniel D. R. Arcanjo, Lúcia F. A. D. Moura, Felipe A. A. Oliveira, Maria J. S. Soares, Patrick V. Quelemes

**Affiliations:** 1Postgraduate Program in Dentistry, Federal University of Piaui, Teresina 64049-550, Brazil; 2Faculty of Medicine, Campus Altamira, Federal University of Pará, Altamira 68372-040, Brazil; 3Laboratory of Functional and Molecular Studies on Physiopharmacology, Department of Biophysics and Physiology, Federal University of Piauí, Teresina 64049-550, Brazil; 4Department of Veterinary Morphophysiology, Federal University of Piauí, Teresina 64049-550, Brazil; 5Medicinal Plants Research Center, Federal University of Piauí, Teresina 64049-550, Brazil

**Keywords:** *Streptococcus* spp., antibiotic resistance, oral cavity, infective endocarditis

## Abstract

Streptococcal bacteremia that occurs during invasive dental procedures can lead to infective endocarditis (IE) in children with certain heart diseases. Prior to such procedures, antibiotic prophylaxis (AP) with amoxicillin (AMPC) is recommended. However, the detection of amoxicillin-resistant strains (AMPC-RS) in the mouths of children with heart diseases raises the concern that they would be uncovered by the action of standard AP. This work carried out a systematic review and meta-analysis regarding AMPC-RS carriage in the mouths of children. We consulted databases covering studies between the years 2000 and 2021, following the PRISMA declaration. A meta-analysis was carried out to assess the prevalence of children carrying AMPC-RS in the mouths. The antimicrobial tests were carried out by microdilution (46.2% of articles), disk diffusion (38.3%), and the E-test (15.4%). *Streptococcus mitis* and *S. sanguinis* were bacteria with the most found resistance phenotype, with MIC reaching values of 128 µg/mL. Of the 13 selected articles, only 6 presented results that made it possible to calculate the prevalence of children carrying AMPC-RS in their mouths, ranging from 5.5% to 86.3%. Most of the studies were classified as high quality, and the collected data demonstrate the presence of streptococcal strains with different levels of resistance in the collected samples, such as the dental plaque. The meta-analysis pointed to evidence of AMPC-RS being carried, with a prevalence of 21.3% (I² = 0%, *p* = 0.705). There is an important prevalence of AMPC-RS carriage in the mouths of children. Specific attention should be directed to AP in those susceptible to IE.

## 1. Introduction

Although viridans group streptococci (VGS) are part of the oral microbiota, they are related to several pathologies [1,2]. Locally, *Streptococcus mutans*, one of its most studied members, is directly associated with the development of dental caries [3]. Systemically, if bacteria from this group enter the bloodstream after invasive dental procedures, for example, they can cause heart infections, such as infective endocarditis (IE), in susceptible patients with certain heart diseases [4,5]. 

The bacteremia caused by such micro-organisms in the risk group described is worrisome because of the adhesins present on the surface of their cell walls, an important virulence factor that helps them to adhere to cardiac prostheses or damaged endocardium, promoting biofilm formation, difficult to eliminate with antibiotics, which can lead to death [4,6].

Currently, it is considered that, in addition to maintaining good oral hygiene, the preventive measure for IE caused by oral bacteria recommended by most international guideline committees is the use of antibiotic therapy prior to invasive dental procedures for susceptible patients [4,7,8,9]. In this context, amoxicillin (AMPC) is still widely used as the first drug of choice for antibiotic prophylaxis (AP) in patients with heart conditions, such as prosthetic valves and congenital valve disease [7,9]. This antibiotic is a broad-spectrum bacteriolytic beta-lactam that inhibits cross-linking between peptidoglycan polymer chains in the cell wall of sensitive bacteria, acting on Gram-positive and Gram-negative bacteria [10].

Considering that IE remains a significant cause of morbidity and mortality, particularly in children with heart disease [11], the AHA (American Heart Association) recommends the use of AP with a dosage of 50 mg/kg of AMPC in suspension, one hour before invasive dental procedures [7,9]. However, due to the excessive and/or indiscriminate use of antibiotics of this class, both for adults and children, AMPC-resistant streptococci (AMPC-RS) have also been detected harboring the oral cavity [12,13,14]. VGS bacteria are considered to show resistance to AMPC (or ampicillin) when this antibiotic has a minimum inhibitory concentration (MIC) > 0.25 µg/mL on such bacteria [15].

Studies still point to disagreements about the presence or prevalence of antibiotic-resistant bacteria of the *Streptococcus* genus in the mouths of systemically healthy children [13,16,17]. When evaluating children with cardiac alterations, a risk group for IE, a higher prevalence of the presence of AMPC-highly resistant streptococci strains in their mouths was found [18], which is a cause for concern, considering that when performing invasive dental procedures in these children, even after the administration of standard AP, such a risk group would be exposed to an ineffective antibiotic [11].

Thus, due to the limited data regarding the surveillance of antibiotic resistance of oral bacteria, this study aimed to carry out a systematic review and meta-analysis of the literature about the AMPC-RS carriage in the mouths of children.

## 2. Results

According to the flowchart proposed by the PRISMA 2020 statement [19], after the selection method, a total of 13 articles were analyzed in this systematic review (Figure 1): 9 from databases and 4 from other search methods. Table 1 summarizes the data collected from the analysis of the selected articles, which will be briefly described as follow.

The studies included are from Japan (23.1%), Kuwait (15.3%), India (7.7%), Kosovo (7.7%), Yemen (7.7%), Poland (7.7%), United Kingdom (7.7%), Iraq (7.7%), Mexico (7.7%), and Brazil (7.7%). According to the method, it can be seen in Table 1 that, of the 13 articles, 9 were considered as having a high quality (>4 points), while 4 articles were classified as having a moderate/low quality (≤4 points). Only one article obtained the maximum score [18].

The collection of samples from the children’s mouths was heterogeneous, with dental biofilm being evaluated in 76.9% (n = 10) [13,17,18,20,21,22,23,24,25,26] of the studies, expectorated saliva in 15.4% (n = 2) [27,28], and the material inside the root canals was analyzed in 7.7% (n = 1) [16]. The strains identification was carried out by using biochemical tests [16,17,21,22,23,26] or molecular biology techniques [13,18,20,24,27,28]. Two of the studies [18,27] (18.2%) sought, in the background, to assess whether the resistant strains found were from transient bacteremia or were part of the oral microflora. The results indicated that the strains did not disappear over approximately 3–4 months in which the interval of collections was submitted, concluding that the strains came from the resident microflora.

For the evaluation of streptococcal antibiotic resistance, the methods applied were agar diffusion tests (38.3% of the articles) [16,17,23,25,26]; E-test (15,4%) [21,22]; and broth microdilution to determine the MIC (46.2%) [13,18,20,24,27,28]. It was found that, in some studies, the most common bacteria related to antimicrobial resistance are not only resistant to a single antibiotic but also to other drugs [18,20,21,22,24,28]. Of the studies that considered the MIC value to investigate the resistance profile, five (38.4%) [13,18,23,27,28] defined the MIC value ≥ 8 µg/mL as a resistance definition parameter.

*S. mitis* was observed with the highest [18,20,22,27] or second highest frequency [13,21] regarding AMPC-resistance. The same way that *S. sanguinis*, with the highest [13,21,23] or second highest [20]. Highly resistant *S. oralis* was found more frequently in a study that verified the endodontic content [24].

Two studies evaluated only *S. mutans* [17,25] and one only *S. salivarius* [28], where a phenotypic pattern of resistance of these micro-organisms was observed not only to beta-lactams but also to other antibiotics, such as the macrolide class, in particular, *S. salivarius*.

**Table 1 pathogens-11-01114-t001:** Data extracted from works included and methodological quality scores.

Reference/ Country Quality Score	Sample Size/ Study Design	Collection/Isolation/Identification Methods	Results	
			Resistance Value Considered (RVC)/ AMPC-RS Found	Number of Children Carrying AMPC-RS
Ready et al. (2004) [20] United Kingdom 1:S; 2:S; 3:N; 4:S; 5:S; 6:S; 7:S; Score: 6	A total of 40 children. Group 1 consisted of 25 who had not used any class of ATB in the 3 months prior to sampling, and group 2 consisted of 15 who had used AMPC in the previous 3 months.	Supragingival dental biofilm removed from tooth surfaces. Calcium alginate swab. Iso-Sensitest with 5% defibrinated horse blood used for the isolation. Identification by using molecular biology techniques.	RVC: ≥8 µg/mL. All the children harbored AMPC-resistant bacteria. Of 224 isolates (MIC range, 8 to 128 μg/mL), 128 AMPC-resistant bacteria were isolated from group 1, and 96 isolated from group 2. The median percentage of the total cultivable oral microbiota resistant to AMPC (mainly *Haemophilus* spp., *Streptococcus* spp. and *Veillonella* spp.) was 2.4% in children without AMPC use and 10.9% in children with AMPC use.	The work does not present this data. However, it points out that the 40 (100%) children carried different strains of bacteria resistant to different ATBs.
Rotimi et al. (2005) [22] Kuwait 1:S; 2:S; 3:N; 4:S; 5:N; 6:S; 7:S; Score: 5	A total of 102 children between 5 and 12 years old of which 88 were from Kuwait and 14 were from other countries.	Supragingival biofilm of deciduous and permanent molars beyond the tongue’s surface. Sterile curettes. Mitis Salivarius agar used for the isolation. Identification by using API 20 Strep test kits (BioMerieux).	RVC: >0.25 µg/mL. In total, 540 strains were isolated. *S. salivarius* was found most frequently (21.5%), followed by *S. sanguis* (16.3%). *S. mitis* (55.9%), *S. oralis* (55.0%), *S. intermedius* (52.3%), *S. anginosus* (45.0%), *S. sanguinis* (45.2%), *S. bovis* (41.7%), *S. salivarius* (39.6%) and *S. mutans* (36.8%) showed high resistance to AMPC.	The work does not present this data.
Salako et al. (2007) [21] Kuwait 1:S; 2:S; 3:N; 4:S; 5:N; 6:S; 7: S; Score: 5	A total of 102 healthy and 102 mentally disabled children between 5 and 12 years old.	Supragingival dental biofilm of all primary and permanent molars. Sterile curettes. Mitis Salivarius agar used for the isolation. Identification by using API 20 Strep test kits (BioMerieux).	RVC: >0.25 µg/mL. In total, 741 strains, 330 from healthy children and 411 from those with disabilities. A higher prevalence of *S. salivarius* in healthy individuals (27.3%) and *S. sanguis* in disabled individuals (22.6%) was found. A high frequency of AMPC-RS, except *S. mutans*, was observed in 18% of healthy children and 21% of children with disabilities.	The work does not present this data.
Nemoto et al. (2011) [13] Japan 1:S; 2:S; 3:N; 4:S; 5:S; 6:S; 7:S; Score: 6	A total of 253 systemically healthy children, teenagers and young adults (2–22 years).	Supra and subgingival biofilm were collected from all teeth. Sterile instrument. Mitis Salivarius agar used for the isolation. Identification by using molecular biology techniques.	RVC: ≥8 µg/mL. Resistant bacteria were selected with selective streptococcal agar with 32 µg/mL of AMPC. In total, 344 strains were isolated; 18 streptococcal strains from 14 patients were highly resistant to AMPC. MICs for these strains ranged from 16 to 64 µg/mL.	A total of 14 children aged between 3 and 9 years.
Nemoto et al. (2013) [18] Japan 1:S; 2:S; 3:S; 4:S; 5:S; 6:S; 7:S; Score: 7	A total of 34 children and adolescents aged 4–19 years with a high risk of developing IE.	Supra and subgingival biofilm was collected from all teeth. Sterile instrument. Mitis Salivarius agar used for the isolation. Identification by using molecular biology techniques.	RVC: ≥8 µg/mL. Resistant bacteria were selected with selective streptococcal agar with 32 µg/mL of AMPC. In total, 9 resistant strains (MIC of 16–64 μg/mL) were found in 7 individuals, each of which was also resistant to other analyzed ATBs, except for new quinolones.	A total of 6 children with AMPC-RS and 1 individual aged 18 years.
Fysal et al. (2013) [16] India 1:S; 2:S; 3:N; 4:S; 5:N; 6:S; 7:S; Score: 5	A total of 50 children aged 6 to 12.	Contents of the oral cavity. Swabs, probes and curettes. Blood agar used for the isolation. Identification by using biochemical tests.	RVC: <20 mm (inhibition halo). In total, 3 strains of *S. mutans* were isolated. All had a diameter > 24 mm, showing no resistance.	The work does not present this data.
Rexhepi et al. (2014) [23] Kosovo 1:S; 2:N; 3:N; 4:S; 5:S; 6:N; 7:N; Score: 4	A total of 90 patients aged 6 to 15 years divided into 3 groups: (n = 30) healthy; (n = 30) with CHA without using ATB in the last 3 months, and (n = 30) with CHA who used ATB in the last 3 months.	Supragingival dental biofilm of dental surfaces. Sterile swab. Nutrient or Blood agar used for the isolation. Identification by using VITEK 2 (BioMerieux) and colorimetric GP card.	The RVC for the research is not mentioned. The disk diffusion method was carried out. In total, *S. mitis* was more present (37.2%), followed by other cocci (8.6%) and then *S. sanguinis* (7.8%). High resistance to AMPC was observed for *S. sanguinis* (20%), then *S. oralis* (13.6%), *S. mitis* (12.9%) and *S. salivarius* (11.1%).	9 children.
Loyola-Rodriguez et al. (2014) [24] Mexico 1:S; 2:S; 3:N; 4:S; 5:N; 6:S; 7:S; Score: 5	A total of 60 children that needed dental treatment for infections and acute symptoms in the primary dentition.	Content inside the canals of deciduous teeth. Endodontic file. BHI agar used for the isolation. Identification by using molecular biology techniques.	RVC: ≥8 µg/mL. Collected samples were inoculated in a culture medium with clindamycin or AMPC at 8 or 16 μg/mL. *S. oralis* and *S mutans* highly resistant to ATBs were found in 75% and 45% of the samples, respectively. The authors do not associate to which ATB these proportions are related.	The work does not present this data.
Nemoto et al. (2015) [27] Japan 1:S; 2:S; 3:N; 4:S; 5:S; 6:S; 7:S; Score: 6	A total of 170 children (4–13 years) and their mothers (150) aged 26 to 49 years, systematically healthy and without ATB use in the last 3 months.	Unstimulated expectorated saliva. Sterile plastic tube. Mitis Salivarius agar used for the isolation. Identification by using molecular biology techniques.	RCV: ≥8 µg/mL. Resistant bacteria were selected with selective streptococcal agar with 32 µg/mL of AMPC. MICs ranged from 16–64 µg/mL. Streptococci highly resistant to AMPC were isolated from 11 children and 7 mothers, which included four mother–child pairs.	11 children.
Palma et al. (2016) [28] Brazil 1:S; 2:S; 3:N; 4:S; 5:S; 6:S; 7:S; Score: 6	A total of 22 oral and systemically healthy infants aged 2 to 16 months were included.	Samples were collected from the oral mucosa. Sterile swab. BHI agar with 5% sheep defibrinated blood and Mitis Salivarius agar were used for the isolation. Identification by using molecular biology techniques.	RCV: ≥4 µg/mL. In total, 95 *S. salivarius* strains were evaluated. High frequencies of infants carrying strains with intermediate resistance to AMPC (n = 16, 72.7%) were found. Among the 95 isolates tested, 75 strains (78.9%) were resistant to at least one ATB among those tested, and 27 (28.4%) were resistant to two or more classes of ATB. MIC of AMPC ranged from 0.03–16 µg/mL.	19 infants.
Krzyściak et al. (2017) [26] Poland 1:S; 2:S; 3:N; 4:S; 5:N; 6:S; 7:N; Score: 4	A total of 143 children with an average age of 4.6 years old. Sample collection in children with and without cavitations due to dental caries.	Dental biofilm on the surface of deciduous molars with early caries. Sterile curettes. HLR-S agar used for the isolation. Identification by using biochemical tests (STREPTOtest, Lachema).	An RCV was not presented for the evaluated ATBs. In total, 142 *S. mutans* strains were divided into four clusters that showed different sensitivity profiles to different ATBs. It was observed that 13% of the isolates were resistant to penicillin, while all were sensitive to vancomycin. The highest MIC of AMPC for *S. mutans* found was 0.5 µg/mL.	The work does not present this data.
Ali Mahmood et al. (2018) [17] Iraq 1:N; 2:N; 3:N; 4:S; 5:N; 6:S; 7:S; Score: 3	A total of 60 dental plaque samples were collected from children aged 3 to 5.	Material collected from caries lesions and supragingival biofilm. Sterile swab. Unclear isolation method. Identification by using biochemical tests.	RVC was not presented for the ATBs evaluated. Of the 120 isolates, 50% were *Streptococcus* spp. The authors tested 100 µL of a 125 mg/5 mL suspension of marketed AMPC from four brands. The diameters of the inhibition zones were observed for *S. mutans*, that were considered slightly susceptible to these drugs.	The work does not present this data.
Al-Shami et al. (2019) [25] Yemen 1:N; 2:N; 3:N; 4:S; 5:N; 6:S; 7:S; Score: 3	A total of 87 biofilm samples from children (2–5 years old) and 87 from their mothers (33–44 years old) with active caries. It was not clear how many mothers and children were evaluated.	Supragingival dental biofilm from sites with active caries. Unclear collection. Mitis Salivarius agar with potassium tellurite, bacitracin and 20% sucrose used for the isolation. Unclear identification method.	An RCV was not presented for the ATBs evaluated by the halo formation method. In total, 174 specimens of *S. mutans* were evaluated. The resistance rate of *S. mutans* to AMPC was 14.9% in isolates from mothers and 12.6% in children.	The work does not present this data.

AMPC-RS (Amoxicillin-resistant streptococci); CHA (congenital heart anomaly); ATB (antibiotic).

Figure 2 presents an overview of the articles that determined the MIC of AMPC for streptococci collected from the oral cavity of children as their main or secondary objective, among other antibiotics and/or bacteria evaluated. This figure shows the bacteria on which the AMPC presented the highest MIC values (and such values) observed in the respective studies.

Considering the admission of early resistance of VGS to AMPC when the MIC reached >0.25 µg/mL [15], the very high values observed in Figure 2 stand out, with MIC reaching values, such as 128 µg/mL [20], a concentration 512 times higher than that considered as the sensitivity standard.

Only 38.5% (n = 5) of the studies had data that allowed us to assess the prevalence of children carrying AMPC-RS in the oral cavity (Table 1) [13,18,23,27,28]. In this context, Figure 3 presents a ratio of the proportion of the total number of children evaluated in relation to those who carried AMPC-RS in their mouths.

One study found AMPC-RS in the mouths of 21% of children and adolescents with congenital heart disease, susceptible to IE [18]. The highest prevalence was found in a study [28] in which 72.7% of the evaluated children (infants) had the bacterium *S. salivarius* with an AMPC resistance phenotype, with MIC ranging from 0.03–16 µg/mL (Table 1).

Figure 4 presents the results of a meta-analysis of the data obtained from selected articles in which it was possible to quantify the number of children carrying AMPC-RS in their mouths, as well as the total number of children evaluated. The forest plot shows a prevalence of children carrying AMPC-RS in their mouths of 21.3% (0.213 with a confidence interval of 0.036–1.267). The calculations showed a low heterogeneity value (I² = 0%, *p* = 0.705), with the prevalence value calculated by the fixed effect model [29].

## 3. Discussion

The presence of AMPC-RS in the oral cavity is already known [12]; however, the harboring of these bacteria in the mouths of children with heart diseases [18] alerted us to the risk that they would suffer when undergoing invasive dental procedures. Thus, to our knowledge, this is the first systematic review that sought and analyzed data from the literature regarding AMPC-RS carriage in the mouths of children.

The method applied in this systematic review was prepared following the PRISMA statement [19], except for the required items not applicable to the scope of this work. The chosen combination of terms was the most effective in the search for articles related to the question. It was necessary to use the Boolean operator “NOT” for the terms “pyogenes” and “pneumoniae”, which define streptococcal species that do not belong to the viridans group [30].

The results showed that the strains were identified by using biochemical or molecular biology methods. The molecular techniques present greater specificity and reliability in relation to identification, especially when trying to understand whether the bacterial samples from the mouth were resident or transient [13,18] or when looking for gene polymorphisms potentially associated with the resistance phenotypes [28].

Although the lack of standardization of microbial susceptibility assessment methods in selected articles can interfere with data interpretation, among the studies, we emphasize that four [13,18,20,27] used a concentration of AMPC for the in vitro selection of resistant streptococci well above the concentration recommended [15].

The methodology applied in some works [13,18,27] used selective agar with 32 µg/mL of AMPC in the search for resistant strains. These authors found a small number of highly resistant strains, for instance, isolating 18 resistant strains from 14 patients out of 253 recruited [13], a relatively small number of isolated strains, compared to the number of participants.

According to international standards, intermediate resistance is considered when a VGS strain has a MIC for AMPC ranging between 0.5 and 4 µg/mL, and complete resistance when it has a MIC ≥ 8 µg/mL [15]. Thus, most likely, if the methodology of the studies mentioned above had used lower concentrations between 2–8 µg/mL of AMPC for isolation, the quantity of resistant strains and the proportion of individuals carrying them would be considerably larger and more alarming.

To assess the quality of the articles and, consequently, their evidence, we applied a standard form used in previous work [31]. We emphasize at the outset that, although some of the evaluated works did not reach a high-quality score, we are aware that their respective objectives were achieved. However, we consider that the items selected to guide us in a coherent quality assessment are relevant to be considered in studies that aim to determine the presence (or prevalence) of resistant micro-organisms in some anatomical site of a given population.

In this manner, we emphasize that the question that deals with verifying whether the results of the work evaluated presented data from which the prevalence of individuals carrying strains with some degree of resistance can be inferred is one of the most important. This data was neglected by most of the reviewed studies since all the articles could have presented that information, important for epidemiological surveillance.

Due to this gap, this review also carried out a meta-analysis of the results obtained with which it was possible to calculate prevalence data. However, not all articles had the data necessary for a deeper statistical verification, such as a comparative meta-analysis of the data necessary to generate more robust evidence sought by the review question. Despite that, an important prevalence of 21.3% of children carrying AMPC-RS in the mouth was calculated. The low heterogeneity of the data obtained made it possible to calculate the prevalence using the fixed effect model, which confers a higher degree of data reliability [29].

The indiscriminate use of antibiotics has caused an increase in the number of resistant strains, also detected harboring the oral cavity [12,13,14]. There are several ways that bacteria avoid the bactericidal effect of beta-lactams, such as an altered penicillin-binding proteins, efflux pumps, and production beta-lactamases [32].

Due to the evidence of the presence and emergence of AMPC-RS in the mouths of children, special attention should be given to those that are susceptible to IE. We highlight that, of the thirteen studies selected, only two evaluated the AMPC susceptibility of oral bacteria collected from children’s mouths with heart conditions, susceptible to IE [18,23]. Therefore, the antibiogram of samples of oral contents for surveillance and choice of effective alternative antibiotic are convenient for this group for an effective AP prior to invasive dental procedures.

## 4. Materials and Methods

### 4.1. Selection of Studies

This systematic review followed the PRISMA statement [19] and sought to answer the following question: What is the prevalence of AMPC-RS carriage in the mouths of children? The CoCoPop mnemonic (condition, context, and population) was applied to defining the mentioned question and the inclusion criteria [33]. The condition was the harboring AMPC-RS; the context considered was that the excessive and/or indiscriminate use of antibiotics can lead to antibiotic resistance of oral bacteria; and the population to be analyzed consisted of children.

Therefore, the MEDLINE via EBSCO, MEDLINE via PubMed, and Web of Science databases were consulted in January 2022 for articles between January 2000 and December 2021. This systematic review was registered in the PROSPERO International Prospective Register of Systematic Reviews database (CRD42022356789).

The terms searched in the Medical Subject Headings (MeSH) were combined as follows for the initial selection of articles: ((amoxicillin) AND (Streptococcus OR streptococci) AND (oral OR mouth) AND (children) NOT (pyogenes) NOT (pneumoniae)). In parallel, we also searched the articles’ references using Google Scholar.

At this stage, three independent researchers (AAGJ, MLVAC, and PVQ) selected the studies and analyzed the titles and abstracts. Disagreements regarding the inclusion or exclusion of works were decided by consensus among the evaluators.

Articles were included in the selection when they were studies that, in their methodology, it was evaluated the susceptibility profile of streptococci collected from the mouths of healthy children or children with co-morbidities to AMPC. During selection, articles were excluded when they referred to research unrelated to the review question or when they were literature reviews, abstracts, case reports, conference proceedings, editorial, or grey literature.

### 4.2. Data Extraction and Quality Assessment of Included Studies

Three evaluators carried out data extraction (AAGJ, MLVAC, and FRPS) and then a fourth carried out verification (PVQ). To this end, we applied the following previously adopted data collection criteria [31]: (a) identification and design of studies—authors, country, year of publication, study period, sample size, and population characteristics; (b) condition assessed at work; (c) collection method—specimen collection instrument and collection sites; (d) bacterial processing, isolation, and identification; and (e) quantity and characteristics of the species, and levels of resistance.

The evaluation of the quality of the articles was carried out by three independent evaluators (AAGJ, MLVAC, and FAAO) and then ratified by two more (PVQ and MJSS), for a consensus, through an adapted checklist [31]. This analysis made it possible to verify the methodological quality used in the selected works, being formulated through seven questions (Table 2), which can be marked as “Yes” when the data are clearly expressed (marking a point) or “No”, in cases of missing data and/or unclear data. The quality of the selected manuscripts was evaluated as high (>4 points) or moderate/low (≤4 points).

### 4.3. Evidence Synthesis and Statistical Analysis (Meta-Analysis)

From the articles in which the presence of AMPC-RS was observed, the bacteria with the highest resistance profile and their respective MICs were grouped in a graphic format. Data from articles in which it was possible to quantify the number of children carrying AMPC-RS in the mouth, as well as the total number of children evaluated, were grouped in a proportion graph format.

With these data, a meta-analysis was carried out to evaluate the prevalence of children harboring AMPC-RS in their mouths, using the Open Meta software version 23.0 (CEBM @Brown), with calculations of the prevalence value with a 95% confidence interval. Data heterogeneity was calculated using the Chi-square test value based on Cochran’s Q test (I2). Both calculations considered values of *p* < 0.05 as significant.

## 5. Conclusions

This systematic review points to evidence of AMPC-RS carriage in the mouths of children, with a prevalence of 21.3%. Most of the articles reviewed were classified as high quality, and the data from their analyses demonstrate the presence of VGS with different levels of resistance to AMPC in the samples collected from oral cavities, especially *S. mitis*, *S. oralis*, *S. sanguinis*, and *S. salivarius*. Therefore, we consider these results as an important contribution to the surveillance of antibiotic resistance of oral bacteria.

## Figures and Tables

**Figure 1 pathogens-11-01114-f001:**
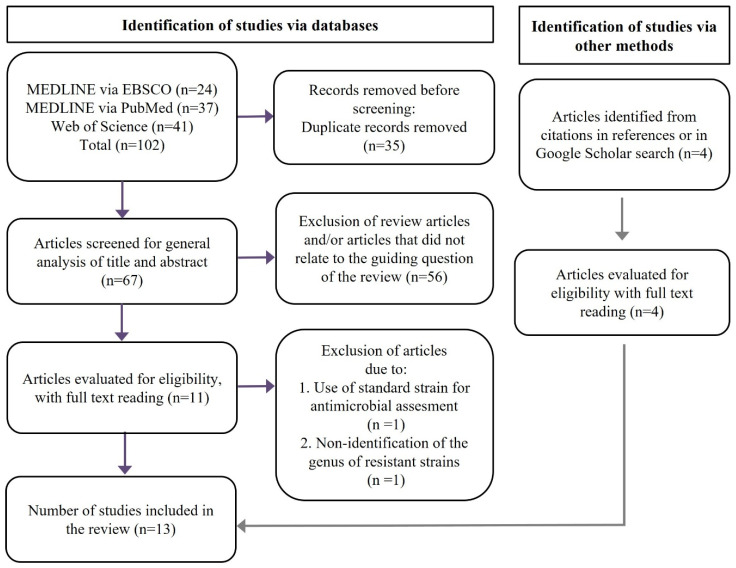
Flowchart of the study selection process for this systematic review.

**Figure 2 pathogens-11-01114-f002:**
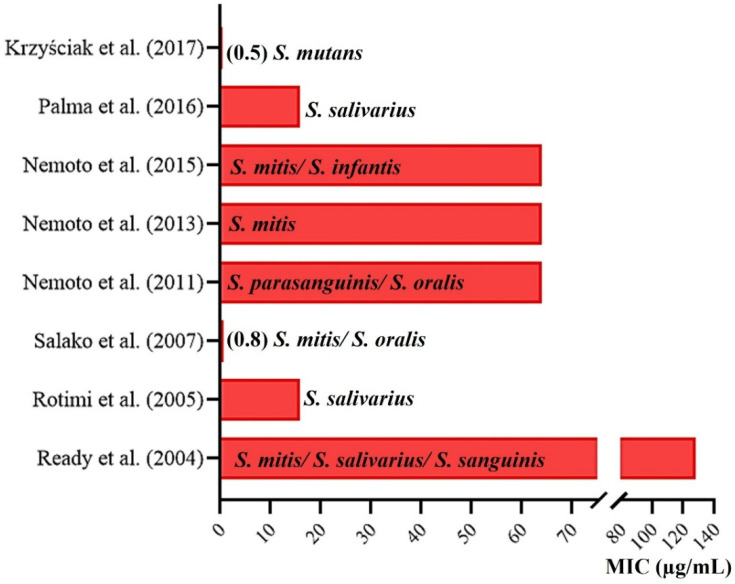
Resistant streptococci that had the highest MIC of AMPC (and their values in µg/mL) found in the reviewed articles [13,18,20,21,22,26,27,28]. Note: intermediate resistance is considered when a VGS strain has a MIC for AMPC ranging between 0.5 and 4 µg/mL, and resistant when it has a MIC ≥ 8 µg/mL [15].

**Figure 3 pathogens-11-01114-f003:**
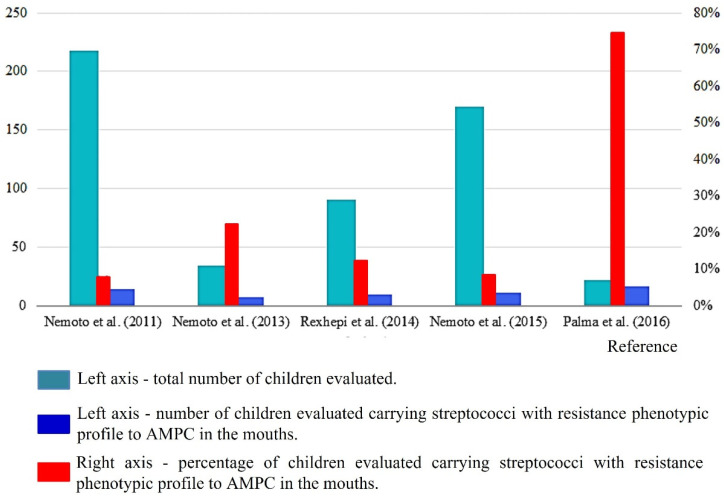
Proportion of the total number of children evaluated in relation to those who carried AMPC-RS in their mouths collected from reviewed studies.

**Figure 4 pathogens-11-01114-f004:**
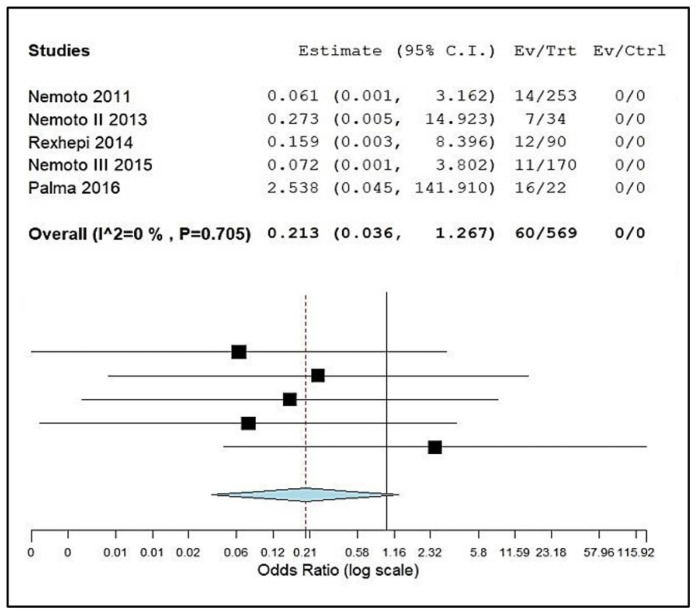
Forest plot of prevalence of children carrying AMPC-RS in their mouths.

**Table 2 pathogens-11-01114-t002:** Criteria for verifying the methodological quality of the included works.

Item	Criteria	Question	Yes	No
01	Population specification	Are the subjects and population studied adequately described?		
02	Collection method	Is there a description of the collection instrument, the selected locations and the form of transport to the analysis?		
03	Sources of bias	Are risks of bias and/or methodological limitations presented and/or discussed?		
04	Information about species of resistant strains	Are resistant strains presented at the species level?		
05	Inference on the epidemiological prevalence	Do the results present data from which the prevalence of individuals carrying strains with some degree of resistance can be inferred?		
06	Controls for reproducibility	Are the methods (MIC *, inhibition halos, biochemical and molecular tests) properly used to determine ATB sensitivity or not?		
07	Interpretation of results	Are antibiotic sensitivity or non-susceptibility standards presented and/or discussed following internationally recognized criteria?		

* MIC—Minimum Inhibitory Concentration.

## Data Availability

Not applicable.

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
