# Peer review of "Amoxicillin-Resistant Streptococci Carriage in the Mouths of Children: A Systematic Review and Meta-Analysis"

_pathogens, 2022, doi:10.3390/pathogens11101114_

Round 1

Reviewer 1 Report

Dear Authors, 

Thank you for the work you submitted. The topic is interesting and the work is well explained, but I feel there should be some changes in the structure. 

- Introduction  is well structured and clear. The last paragraph regarding the aim should be probably expanded; e.g. why would the results be useful? How are you going to research the topic and why? 

- I do not understand why Material and methods section is explained after the results one. It should be right after  the Introduction section, and after that Results should be explained.

 Further, Discussion and Conclusion sections should be described.

Furthermore, the PRISMA flow chart should be included in the Material and Methods section.

Please, focuse on the proper order and structure of the manuscript. 

- Conclusions does not give a clear explanation on how the results can be a useful addition and contribution to literature and to the clinical practice. 

I believe once these issues will be overcome, the manuscript could be suitable for publication.

Best Regards

Author Response

Response to Reviewer 1 Comments

Dear Authors,

Responses - Dear reviewer, thank you for your comments. We will answer them point by point below:

Thank you for the work you submitted. The topic is interesting and the work is well explained, but I feel there should be some changes in the structure.

- Introduction is well structured and clear. The last paragraph regarding the aim should be probably expanded; e.g. why would the results be useful? How are you going to research the topic and why?

Responses - Dear reviewer, thank you for your suggestion. We have added more details of the importance of the work, however, some of your questions have been addressed in Discussion Section.

- I do not understand why Material and methods section is explained after the results one. It should be right after the Introduction section, and after that Results should be explained. Further, Discussion and Conclusion sections should be described.

Responses - Dear reviewer, the order of topics is a requirement of the journal Pathogens.

Furthermore, the PRISMA flow chart should be included in the Material and Methods section.

Responses - Dear reviewer, we understand that some results obtained by applying the proposed method in this Systematic Review can be found in PRISMA flow chart, for this reason we have entered it in the Results Section.

Please, focuse on the proper order and structure of the manuscript.

Responses - Dear reviewer, we followed the order of the structure proposed by the journal's template.

Conclusions does not give a clear explanation on how the results can be a useful addition and contribution to literature and to the clinical practice.

Responses - Dear reviewer, the aspects pointed out are described in the Discussion Section. In our conclusion, we objectively answered the proposed objective.

I believe once these issues will be overcome, the manuscript could be suitable for publication.

Best Regards

Reviewer 2 Report

Authors summarized regarding amoxicillin-resistant streptococci carriage in the mouse of children in this systematic review. I have several suspicious points to the structure of this article. Please answer against my comments and questions, and revise your manuscript.

1.       Did you collect the epidemiological data of children with IE caused by AMPC-RS? I just found a reference no.18. This case seems like a minor case, that is, it is too small to make the topic as systematic review. You should add more the evidence as references in the section of introduction.

2.       Authors selected the articles according to the guideline of PRISMA check list. However, as the results of Table 1, how distinguish between AMPC and the natural resistance of amoxicillin? The sample design contains many healthy children in the Table 1.

3.       Add the “Isolation and Identification” of Streptococci to Table 1, and add the discussion about them.

4.       Explain the difference of “harboring” and “carrying” in Table 3.

5.       Register the PROSPERO before the next submission, and show the number.

Author Response

Response to Reviewer 2 Comments

Dear reviewer, thank you for your comments. We will answer them point by point below:

Authors summarized regarding amoxicillin-resistant streptococci carriage in the mouth of children in this systematic review. I have several suspicious points to the structure of this article. Please answer against my comments and questions and revise your manuscript.

  1. Did you collect the epidemiological data of children with IE caused by AMPC-RS? I just found a reference no.18. This case seems like a minor case, that is, it is too small to make the topic as systematic review. You should add more the evidence as references in the section of introduction.

Dear reviewer, this systematic review sought to answer the following question: What is the prevalence of AMPC-RS carriage in the mouth of children?

In this context, we emphasize that, to the best of our knowledge, this is the first systematic review about this approach, very important to the epidemiological surveillance of antibiotic resistance of oral bacteria.

The applied method resulted in 13 works to be analyzed. We believe that the small number of articles shows that the topic deserves better attention.

We did not aim to review whether amoxicillin-resistant streptococci can cause IE. In Introduction Section are references that point to this possibility:

Baltimore, R.S.; Gewitz, M.; Baddour, L.M.; Beerman, L. B.; Jackson, M. A.; Lockhart, P. B.; Pahl, E.; Schutze, G. E.; Shulman, S. T.; Willoughby, R. Infective Endocarditis in Childhood: 2015 Update - A Scientific Statement from the American Heart Association. Circulation 2015, 132 (15), 1487-1515; doi: 10.1161/CIR.0000000000000298

Wilson, W. R., Gewitz, M., Lockhart, P. B., Bolger, A. F., DeSimone, D. C., Kazi, D. S., Couper, D. J., Beaton, A., Kilmartin, C., Miro, J. M., Sable, C., Jackson, M. A., Baddour, L. M. Prevention of Viridans Group Streptococcal Infective Endocarditis: A Scientific Statement From the American Heart Association. Circulation. 2021, 143(20); 963-978; doi: 10.1161/CIR.0000000000000969

In our work, we have highlighted the risk to infective IE in children with certain heart diseases harboring AMPC-RS, group in which prophylactic antibiotic therapy with AMPC is recommended. Due to this reason, we have suggested in Discussion Section the antibiogram of samples of oral contents for surveillance and choice of effective alternative antibiotic for this group for an effective AP prior to invasive dental procedures.

Regarding the healthy children, we believe that our results contribute to the epidemiological surveillance of antibiotic resistance in oral bacteria.

Therefore, our results pointed to an important prevalence that may be being neglected.

  1. Authors selected the articles according to the guideline of PRISMA check list. However, as the results of Table 1, how distinguish between AMPC and the natural resistance of amoxicillin? The sample design contains many healthy children in the Table 1.

Dear reviewer, we did not find literature data that evidence natural (intrinsic) resistance of viridans group streptococci to amoxicillin.

Most children participating in the reviewed studies did not have systemic medical conditions. Due to your consideration, we have added a topic to the discussion regarding the lack of studies with children with heart disease.

  1. Add the “Isolation and Identification” of Streptococci to Table 1 and add the discussion about them.

Dear reviewer, thank you for your suggestion. We have added.

  1. Explain the difference of “harboring” and “carrying” in Table 3.

Dear reviewer, in this context, these words are usually used synonymously. However, we carried out a correction in Figure 3. Thank you.

  1. Register the PROSPERO before the next submission and show the number.

Dear reviewer, thank you for your observation: CRD42022356789

Round 2

Reviewer 2 Report

Thank you for your revised manuscript.

According to your answers for my questions, I understood the purpose of this systematic review.

The revised manuscript was improved, and answers for my comments were sufficient.